# Prediction of 30-Day Readmission in Hospitalized Older Adults Using Comprehensive Geriatric Assessment and LACE Index and HOSPITAL Score

**DOI:** 10.3390/ijerph20010348

**Published:** 2022-12-26

**Authors:** Chia-Hui Sun, Yin-Yi Chou, Yu-Shan Lee, Shuo-Chun Weng, Cheng-Fu Lin, Fu-Hsuan Kuo, Pi-Shan Hsu, Shih-Yi Lin

**Affiliations:** 1Department of Family Medicine, Taichung Veterans General Hospital, Taichung 40705, Taiwan; 2Center for Geriatrics & Gerontology, Taichung Veterans General Hospital, Taichung 40705, Taiwan; 3Division of Allergy, Immunology and Rheumatology, Department of Internal Medicine, Taichung Veterans General Hospital, Taichung 40705, Taiwan; 4Department of Neurology, Neurological Institute, Taichung Veterans General Hospital, Taichung 40705, Taiwan; 5Division of Nephrology, Department of Internal Medicine, Taichung Veterans General Hospital, Taichung 40705, Taiwan; 6Division of Occupational Medicine, Department of Emergency, Taichung Veterans General Hospital, Taichung 40705, Taiwan; 7Division of Endocrinology and Metabolism, Department of Internal Medicine, Taichung Veterans General Hospital, Taichung 40705, Taiwan

**Keywords:** hospital readmission, elders, comprehensive geriatric assessment, LACE index, HOSPITAL score

## Abstract

(1) Background: Elders have higher rates of rehospitalization, especially those with functional decline. We aimed to investigate potential predictors of 30-day readmission risk by comprehensive geriatric assessment (CGA) in hospitalized patients aged 65 years or older and to examine the predictive ability of the LACE index and HOSPITAL score in older patients with a combination of malnutrition and physical dysfunction. (2) Methods: We included patients admitted to a geriatric ward in a tertiary hospital from July 2012 to August 2018. CGA components including cognitive, functional, nutritional, and social parameters were assessed at admission and recorded, as well as clinical information. The association factors with 30-day hospital readmission were analyzed by multivariate logistic regression analysis. The predictive ability of the LACE and HOSPITAL score was assessed using receiver operator characteristic curve analysis. (3) Results: During the study period, 1509 patients admitted to a ward were recorded. Of these patients, 233 (15.4%) were readmitted within 30 days. Those who were readmitted presented with higher comorbidity numbers and poorer performance of CGA, including gait ability, activities of daily living (ADL), and nutritional status. Multivariate regression analysis showed that male gender and moderately impaired gait ability were independently correlated with 30-day hospital readmissions, while other components such as functional impairment (as ADL) and nutritional status were not associated with 30-day rehospitalization. The receiver operating characteristics for the LACE index and HOSPITAL score showed that both predicting scores performed poorly at predicting 30-day hospital readmission (C-statistic = 0.59) and did not perform better in any of the subgroups. (4) Conclusions: Our study showed that only some components of CGA, mobile disability, and gender were independently associated with increased risk of readmission. However, the LACE index and HOSPITAL score had a poor discriminating ability for predicting 30-day hospitalization in all and subgroup patients. Further identifiers are required to better estimate the 30-day readmission rates in this patient population.

## 1. Introduction

Hospital readmissions are associated with unfavorable patient outcomes and high financial burden. Considerable research has been conducted on this issue worldwide [1]. Importantly, older individuals may experience a higher risk of repeated admissions due to more comorbidities, declined physical and cognitive functional status, and less socioeconomic support [2].

Taiwan’s population, one of the most rapidly aging worldwide, reached the threshold that defines a society as aged in April 2018, and is expected to become a super-aged society (20% of population consisting of the elderly) by 2030 [3]. As older people can have higher utilization of healthcare and support services [4,5], particularly involving more frequent hospitalization [6], it is important for medical staff in Taiwan to identify older patients at high risk of readmissions as early as possible to improve care quality and reduce medical costs.

Comprehensive geriatric assessment (CGA) has been shown to predict clinical outcomes in older populations [7]. The central concept underlying CGA involves a coordinated and multidimensional diagnostic approach, which helps assess an older patient’s medical, functional, mental, and social condition [8], and therefore, these integrated variables could be used as predictors of important outcomes, such as hospital readmissions and death. For example, activities of daily living have been implemented as predictors of 30-day unplanned readmission and death [9,10]. Moreover, malnutrition, frailty, and polypharmacy were associated with hospital readmissions. CGA-based interventions have also been proven to reduce mortality and institutionalization in patients admitted to an acute geriatric ward [7,11,12,13,14].

Several prediction models of readmission have been introduced [15], with two of the most cited risk assessment tools being the LACE index, which is calculated using four items, namely length of stay, acuity of admission, Charlson Comorbidity Index, emergency department [ED] visits in the past 6 months); and the HOSPITAL score, which consists of seven readily available clinical predictors, namely serum sodium, hemoglobin, length of stay, procedure during hospital, previous admission numbers, index admission type, and discharge from oncology service [16]. Owing to their simple features and high replicability, they have been applied in various clinical settings worldwide [17]. However, assessments of their performance have been hindered by the widely differing populations that have been investigated. Among middle-aged patients free of serious comorbidities, the LACE index was reported to have only a fair discriminative ability in identifying young individuals at higher risk of 30-day readmissions [18]. Furthermore, in older patients, the performance of the LACE index was reported to be worse [19].

To date, there have been few studies to explore the role of the LACE index and HOSPITAL scores, particularly in older patients with physical function impairment and malnutrition for predicting 30-day unplanned readmission in an older adult population. This has particular significance for the older populations (e.g., acute geriatric ward) as they tend to have complex related conditions.

The objective of this study was to investigate the potential predictors of 30-day readmission risk by CGA in patients aged 65 years or older admitted to the geriatric ward of a large academic medical center mainly to receive acute care. In addition, we examined the predictive ability of the LACE index and HOSPITAL score, particularly in older patients with malnutrition and physical dysfunction, both of which are known risk factors for readmission among older adults.

## 2. Materials and Methods

The study explored the clinically relevant factors affecting 30-day hospital readmissions in elderly patients admitted to an acute geriatric ward. In this study, the standard methodology was used to report findings according to the ‘STrengthening the Reporting of OBservational studies in Epidemiology’ (STROBE) guidelines [20]. Because all data were based on the patient-registered health system geriatric assessment and care database of Taichung Veterans General Hospital, and were analyzed anonymously in a retrospective manner, verbal or written consent was not required from the enrolled subjects, according to the regulations of our hospital’s ethics committee. The study was approved following an ethical review conducted by the Institutional Review Board of Taichung Veterans General Hospital (IRB No. CE18141B).

Patients aged 65 or older who were admitted to the acute geriatric ward of a medical center in Taiwan between July 2012 and August 2018 were enrolled. We examined 2106 patients in total. The patients were all admitted to our ward due to acute illness, and the top leading causes of admission among our population were pneumonia, urinary tract infection and cellulitis, and the most common comorbidities were hypertension, diabetes, and chronic obstructive pulmonary disease according to our previous report [21,22]. Discharge was arranged after recovery of acute illness such as infection. None of the patients were discharged against medical advice. Patients were excluded if they died before discharge, had limited ability to receive comprehensive geriatric assessment, or had incomplete data on components of the LACE index or HOSPITAL score; 597 patients were excluded from the study. Finally, 1509 patients were included for analysis.

Patients’ general demographic data were obtained from patients’ electronic medical records, and included age, sex, emergency room visits, admission in the past 6 months, education level, marital status, and socioeconomic status. We also assessed patients’ medical history and comorbidities for any diagnosed diseases and medications and calculated the age-adjusted Charlson Comorbidity Index (ACCI) scores for each patient. At both admission and discharge, all participants were evaluated by a comprehensive geriatric assessment (CGA), which included the Mini-Mental State Examination (MMSE), Mini-Cog, 5-item Geriatric Depression Scale (GDS-5), Activities of Daily Living (ADL), Instrumental Activities of Daily Living (IADL), Mini-Nutritional Assessment (MNA), Hand Grip Strength (HGS), 6-m Walking Speed (6M), and Timed Up and Go test (TUG). The brief descriptions of instruments of CGA are shown in Table 1. These evaluations were conducted by well-trained physicians and nurses. Cognitive status was measured using the Mini-Cog and Mini-Mental State Examination (MMSE) [23]. The normal and abnormal cognitive function cut-off points, as defined by the Taiwanese Mini-Mental State Examination (T-MMSE), were adjusted based on age and educational level [24,25]. Mood was screened using the 5-item Geriatric Depression Scale (GDS-5) [26]. Polypharmacy was defined as more than 5 drugs used daily in our cohort. Nutritional status was assessed using Mini-Nutritional Assessment (MNA) scoring [27,28]. Frailty was assessed by handgrip strength (HGS) using a handheld dynamometer (Smedley’s Dynamometer, TTM, Tokyo, Japan), walking speed (meter/s) on a 6 m course (6M), and Timed Up and Go test (TUG) [29,30,31]. Gait ability at baseline was assessed subjectively by the patient on a scale of 0 (able to walk independently) to 4 (incapable of walking at all). Physical function was assessed by the Barthel Index (BI) for Activities of Daily Living (ADL), and Lawton–Brody scale for Instrumental Activities of Daily Living (IADL) [32,33]. Patients’ ADL and IADL were recorded at baseline, at admission, and at discharge.

The LACE index and HOSPITAL score were calculated for each patient. The LACE index consists of length of hospital stay (“L”), acuity of the admission (“A”), comorbidities of patients (“C”, based on the Charlson Comorbidity Index), and emergency visits within 6 months (“E”), and ranges from 0 to 19 [34]. The original HOSPITAL score developed by Donze et al. in 2013 was used in this study [35], but the number of hospital admissions during the previous year was adjusted to the number of hospital admissions in the past 6 months. All patients were deemed as having an ICD-9-coded procedure during their hospital stay, and none of the patients were “discharged from an oncology service” due to the characteristics of the acute geriatric ward.

**Table 1 ijerph-20-00348-t001:** CGA instruments and interpretations.

	Cut-Off and Interpretation
Mini-Mental State Examination (MMSE)	22-item questionnaire on time and place orientation, registration, attention and calculation, recall, language, repetition, and constructional ability, with total score of 30. Patients are considered with cognitive impairment if ≤24 in the literate and ≤13 in the illiterate [23,25].
Mini-Cog	Two-part exam on memory and orientation—three-item recall and clock drawing test (CDT) [36,37]. Patients are considered demented if there are 0 items recalled or clock drawing test is abnormal with 1–2 items being recalled.
5-item Geriatric Depression Scale (GDS-5)	5-item questionnaire on depressive symptoms with total score of 0–5 [26,38,39]. Patients are considered with depressive symptoms if points ≤2 and psychiatric consultation is suggested.
Barthel Index (BI) for Activities of Daily Living (ADL)	10-item questionnaire on one’s ability of feeding, bathing, grooming, dressing, bowel control, bladder control, toilet use, transferring, and mobility on level surfaces and stairs [32]; total score of 0–100, from total dependence to complete independence.
Lawton–Brody scale for Instrumental Activities of Daily Living (IADL)	8-item questionnaire on one’s ability of using a telephone, laundry, shopping, preparing food, housekeeping, using transportation, managing one’s own medication, and handling finance [33,40]; total score 0–8, from lower physical and cognitive function to higher function.
Mini-Nutritional Assessment (MNA)	30-point questionnaire on dietary and nutritional status, with total score 0–30. Patients are considered with adequate nutritional status if scored ≥24 and malnutrition if scored <17; there is higher risk of malnutrition if scored between 17 and 23.5 [27,41].
6 m Walking Speed (6M)	Gait speed less than 1 m/s indicated low physical performance [42].
Timed Up and Go test (TUG)	Time measured from standing up from a chair, walking for 3 m, returning to the chair, and sitting. Frailty and gait impairment is considered if the patient took more than 30 s to complete the test, and gait aid is required [31].

The overall outcome was 30-day unscheduled admission, defined as a hospital admission into TCVGH for any diagnosis within 30 days of discharge from the index admission at the acute geriatric ward. These data were collected from the hospital’s electronic medical records and analyzed.

Continuous variables were expressed as median and interquartile range (IQR, 25–75%). Categorical data were expressed as number and percentage of the total number of participants. Due to skewed distribution of continuous data, comparisons were made using Mann–Whitney U test for continuous variables and chi-square test for categorical variables. Univariate logistic regression analysis was used to examine all potential factors associated with 30-day hospital readmissions (e.g., age, gender, demographic factors, LACE index and HOSPITAL scores, and comprehensive geriatric assessment variables). Then, multivariable logistic regression was conducted if the variables were statistically significant in the univariable logistic regression. To assess the discrimination ability of LACE index and HOSPITAL scores to predict 30-day hospital readmission among our study cohort, we also conducted receiver operating characteristic (ROC) analysis; the discriminative ability of a risk prediction model was expressed as C-statistic, which ranges from 0.5 (no discrimination, no better than chance) to 1.0 (perfect discrimination). The best cut-off values of the LACE index and HOSPITAL scores were determined by the ROC curve. Moreover, ROC analysis for predicting the ability of LACE and HOSPITAL scores for 30-day hospital readmissions was performed in subgroups according to the components of CGA showing an association with readmission. Statistical analyses were performed using SPSS version 22.0 (SPSS, Chicago, IL, USA). The level of statistical significance was set at *p* < 0.05.

## 3. Results

Table 2 shows that the median age of the study population was 82 years, with the majority being male patients (61.56%, *n* = 929). Most patients were admitted through the emergency room (63.22%, *n* = 954), while the remainder were admitted through the outpatient department. The median length of stay was 10 days, and the 30-day readmission rate was 15.4%.

### 3.1. Baseline Characteristics, CGA, and 30-Day Readmission

A total of 233 out of the 1509 patients were readmitted within 30 days. As shown in Table 1, they had a longer length of stay and more emergency department visits and admissions within 6 months as compared to the non-admitted patients. Moreover, they presented with significantly higher ACCI, representing more severe comorbidity. Regarding the comprehensive geriatric assessment (CGA), patients readmitted within 30 days presented with poorer performance in gait ability, activities of daily living, and instrumental activities of daily living, both upon admission and at discharge. In addition, they were more likely to present with malnutrition, as evidenced by their Mini-Nutrition Assessment (MNA) scores.

Univariable and multivariable logistic regressions were performed to determine the relevant factors related to 30-day readmissions, and the results are shown in Table 3. LACE index and HOSPITAL score were included in multivariable regression analysis, respectively. Male gender, as well as impairments in ADL, IADL, MNA, and gait ability, were significantly associated with 30-day readmissions according to the univariable regression analysis. After adjustment for the other covariates, multivariable logistic regression showed that apart from the LACE index and HOSPITAL score, male gender and moderate impairment in gait ability were independently and significantly correlated with 30-day hospital readmissions.

### 3.2. LACE Index and HOSPITAL Score and 30-Day Readmission

The mean LACE index and HOSPITAL score examined upon admission were 11.1 and 4.42, respectively. In the patients with 30-day readmission, the LACE index and HOSPITAL score were significantly higher than in the non-readmission individuals. The LACE index and HOSPITAL score were shown to be significantly associated with 30-day readmission using univariable regression analysis. The multivariable logistic regression also showed that both LACE index and HOSPITAL score were strong significant predictors for 30-day hospital readmissions. Furthermore, we conducted receiver operator characteristic (ROC) curve analysis to determine the discrimination ability of the LACE index and HOSPITAL score for predicting 30-day readmission, as shown in Figure 1, and showed an area under the curve, or C-statistics of 0.59 for both indexes, indicating poor discriminative ability. Table 4 shows an overview of the sensitivity, specificity, for each cutoff point between high and low risk patients for 30-day readmission. The model had optimal balance between sensitivity and specificity for predicting 30-day all-cause unplanned readmission at a LACE index ≥16 And HOPSITAL score ≥ 6. This threshold (cut-off point) yielded a sensitivity of 8.6% and specificity of 95.8% for LACE index, and a sensitivity of 15.0% and specificity of 91.2% for HOSPITAL score, with an AUC of 0.59 for both.

### 3.3. LACE Index and HOSPITAL Score in Subgroup Patients with Physical Limitation and Malnutrition and 30-Day Readmission

We then performed ROC analysis for the predictive ability of the LACE and HOSPITAL score for 30-day hospital readmissions in subgroups, as shown in Table 5. These included the subgroups of age 85 years and above, age below 85 years, sex (male or female), MNA 24 and above, MNA below 24, ADL upon admission 60 and above, ADL upon admission below 60, fair gait ability, and mildly decreased gait ability. The C-statistics of the LACE index and HOSPITAL score for these subgroups ranged from 0.55 to 0.62, which showed that the LACE and HOSPITAL scores still had poor discrimination ability for 30-day readmissions among subgroups.

## 4. Discussion

Our study found that functional status, nutritional status, gait ability, and multimorbidity were associated with readmission, and after adjustment for potential confounders, it was shown that moderate impairment in gait ability remained significantly predictive of hospital readmission. Furthermore, both the LACE index and HOSPITAL score had a low discriminating ability for predicting readmission in 30 days, with a C-statistic of 0.59. The discrimination ability did not improve when applied in subgroups with abnormal results of CGA.

### 4.1. CGA and Readmission

In this study, we searched for components of the comprehensive geriatric assessment (CGA) that correlated with hospital readmission. Our results showed that although patients with 30-day hospital readmissions suffered from more severe impairment in gait ability, functional impairment, and malnutrition, impairment of these CGA components had a limited ability to predict 30-day hospital readmissions in our population. Prior studies had shown that there was a higher readmission rate, either early or late, among malnourished hospitalized patients as compared to well-nourished patients [43]. In another nationwide study of hospital admissions among seniors, functional impairments were associated with higher readmission rates, with a trend of rising odds of readmission with increasing severity of functional impairment [10]. In contrast, a prospective cohort study of patients above 75 years reported that malnutrition and loss of independence in ADL were associated only with risk of death but not hospital readmissions [12]. Several cohort studies have reported a strong association between polypharmacy and important clinical outcomes such as potentially inappropriate medications (PIMs), hospital readmission, and mortality [44,45,46]. In our study, the association between polypharmacy and 30-day hospital readmission was not significant. This could have resulted from a high prevalence of polypharmacy related to multimorbidity, malnutrition, and physical dysfunction in our cohort of elders [47].

The Multidimensional Prognostic Index (MPI), a prognostic tool in CGA, showed good discrimination and calibration for mortality of the elderly [48]. Based on a large prospective observational study by Pilotto et al., MPI showed only fair discrimination for predicting hospital readmissions, with a C-statistic of 0.65 [49]; and in a cohort study by Hansen et al. it was reported that MPI had poorer discrimination for readmission compared to mortality, with a C-statistic of 0.59 for 30-day readmission and 0.76 for 90-day mortality [50]. It was speculated that functional decline and malnutrition might play less of a role in the short-term readmission of elderly patients after discharge compared to short-term or long-term mortality. However, since repeat hospitalization may exacerbate malnutrition, and functional dependence was associated with increased healthcare expenditures and mortality, it is still essential to assess these components during hospitalization.

Gait impairment has also shown a correlation with hospital readmission in our study. Decreased walking speed has been proposed to be a surrogate marker of frailty and is associated with mortality, hospitalization, and nursing home disposition [51]. Further, some previous studies reported that low gait speed at discharge was associated with an increased risk of readmission [51,52]. It is hypothesized that gait impairment may represent a composite of other risk factors such as age, comorbidities, poor cardiopulmonary function, decrease in muscle strength or mass [53], falls [54], and the other characteristics that may be related to the risk of early readmission. However, in our study patients, the exact mechanisms linking gait impairment and readmission still need further investigation.

Our study also found that male gender was significantly associated with hospital readmission in our cohort. Reports have suggested that in older men with adult children as caregivers, there is a strong correlation with medication-related hospital readmissions [55]. Moreover, nonadherence to medication prescribed by physicians has been reported in previous studies, and it was found that men are more likely to forget or change the dosage themselves [56]. Conflicting results have been reported, however, for gender-associated hospital readmission [11,44]. As this issue is multifactorial (e.g., polypharmacy, nonadherence, multimorbidity, cultural, and ethical considerations), further analysis is required to clarify the association between sex and hospital readmission.

### 4.2. LACE/HOSPITAL and Readmission

Among multiple prediction models for readmission, the LACE index and HOSPITAL score were most widely used, as they are simple and have been validated in multiple countries. However, their performance for predicting 30-day hospital readmission varied widely among different populations. When first developed, the LACE index showed a C-statistic of 0.684 among medical and surgical patients, and HOSPITAL score with a C-statistic of 0.67 among patients admitted in medical service [34,35]. According to Heppleston et al., the LACE index predicts hospital readmission among all ages, which is more pronounced among younger individuals [18,57], but in two cohorts of elderly patients in the UK [19] and Singapore [1], the LACE index performed poorly. The poor performance of the LACE index was consistent with our study, as the mean age of our cohort was 80 years, which was much older than the mean age of the population, which the LACE index and HOSPITAL score were originally derived from, which was 61.3 years. In addition to the difference in mean age, our study population also had a longer length of stay (median of 5 days vs. 10 days, 84.76% had >5 days of LoS), much higher frequency of emergency department visits and hospital admission in 6 months prior to the index admission, and higher Charlson Comorbidity Index scores (mean of 0.5 in LACE index vs. 2.58) in comparison with the original validation study [34]. Furthermore, all patients in our geriatric ward received ICD-9-coded procedures, and none of them were admitted to an oncology service due to the specific characteristics of our ward. Taken together, due to significant differences in study populations, the performance of the LACE index and HOSPITAL score was not particularly discriminative.

As a component of both the LACE index and HOSPITAL score, length of stay alone was associated with 30-day readmission in our cohort as well. While longer length of stay may indicate optimization of therapy [58], it may also expose patients to potential nosocomial infections, disrupted sleep, delirium, or cost burden, which increase the risk of hospital readmission [58,59]. Moreover, longer length of stay could also represent more severe illness or comorbidities, which is typical for elderly patients in hospital; such conditions simply require longer admission and pose a higher risk of readmission [7].

In addition to length of stay, multimorbidity, represented by the age-adjusted Charlson Comorbidity Index, was independently associated with 30-day hospital readmission in our cohort. This was in line with past studies, and multiple observational studies showed that multimorbidity is associated with not only hospital readmission [60], but also increased specialized care, referral, and more prescriptions, thereby increasing the risk of polypharmacy and potentially inappropriate medications (PIMs) [61,62,63]. Two systematic reviews also showed a positive association of multimorbidity and in-hospital mortality and healthcare utilization/cost outcomes [64,65]. Due to its effect on both individuals and healthcare systems, emerging interventions have been carried out, though the effectiveness of each intervention remains uncertain [66].

Numerous studies have been conducted on the performance of the LACE index and HOSPITAL score for 30-day readmission risk prediction in some specific disease settings, including heart failure, plastic surgery, COPD, and pneumonia with variable predictive ability [67,68,69,70]. In this study, we further examined the discrimination ability of these two risk scores in older hospitalized patients with malnutrition and physical dysfunction. It was shown that the performance of the LACE index and HOSPITAL score at predicting hospital readmissions were not satisfactory. It has been speculated that due to the wide range of complex contributing factors affecting hospital readmission, it may not be possible to implement a simple predictive score for patient populations [71].

Although the LACE index and HOSPITAL score showed poor discriminating ability in predicting 30-day readmission in our study population of elderly patients in Taiwan, there is still a dire need for a prediction tool for hospital readmission for elderly patients, particularly as Taiwan is set to become a super-aged society. By identifying patients with higher risk, further interventions, such as early rehabilitation programs, more specialized care, comprehensive discharge planning and patient-centered education during admission, timely postdischarge follow-up, and communication with primary care providers, could be implemented in advance, thereby improving the quality of care and lightening the burden on the healthcare system [62,72,73,74,75].

After identification of impairment or frailty by implementation of CGA, interventions are required to reduce further deterioration of frailty and to prevent future complications arising from frailty or geriatric syndrome [76,77,78]. The interventions mostly include exercise, nutritional intervention, or both [79]. A personalized intervention for each individual is favored over standardized management for all frail elderly [80], depending on one’s type of frailty, socioeconomic and environmental factors, and available resources. However, in our study, multidimensional interventions were carried out based on CGA, and effects of individual interventions were not analyzed in this study. Furthermore, in spite of several benefits of CGA for managements in hospitalized patients [8], it should be noted that CGA is time-consuming and requires a geriatrician, and the assessment process and further treatment plan have been shown to be difficult to standardize [81]. Several modified versions of CGA, such as rapid geriatric assessment (RGA) and geriatric assessment for oncology, have been developed to evaluate older patients for geriatric syndromes more quickly, and may be utilized among non-geriatricians and primary care practitioners [82,83,84]. Future studies could be focused on the feasibility of modified geriatric assessment and which intervention may benefit older hospitalized patients more by rapid or short CGA.

### 4.3. Strength and Limitations

The strengths of our study were the long study duration (7 years) with a moderately sized cohort, with well-constructed measurements of comprehensive geriatric assessment of patients, including baseline status, status upon admission, and status upon discharge. Still, we acknowledge some limitations in our study. First, this study had relatively low sensitivity based on the scoring system with moderate C-statistics. Due to the characteristics of the acute geriatric ward, there was no oncology admitting service, and no complete records of ICD-9-coded procedures. Validation of predictive models would be biased, and thus sensitivity would be reduced. Second, only readmissions to the same hospital were considered. Our patients might have been readmitted to other hospitals after discharge, and therefore, the readmission rate would have been underestimated. Third, our study did not consider factors such as social and family supportive factors in the comprehensive geriatric assessment, both of which may have influenced the readmission risk and predictive ability of the tools used. Finally, the included patients came from a single ward in a single hospital, and this may have limited the applicability of the results.

## 5. Conclusions

Our study showed that only some components of CGA, mobile disability, and gender were independently associated with increased risk of readmission. Moreover, the internationally validated LACE index and HOSPITAL score, although both were independently associated with 30-day readmission risk, did not have adequate discriminative capability either in all patients or subgroup patients with physical disability and malnutrition. Further identifiers are required to better estimate the 30-day readmission rates in this patient population.

## Figures and Tables

**Figure 1 ijerph-20-00348-f001:**
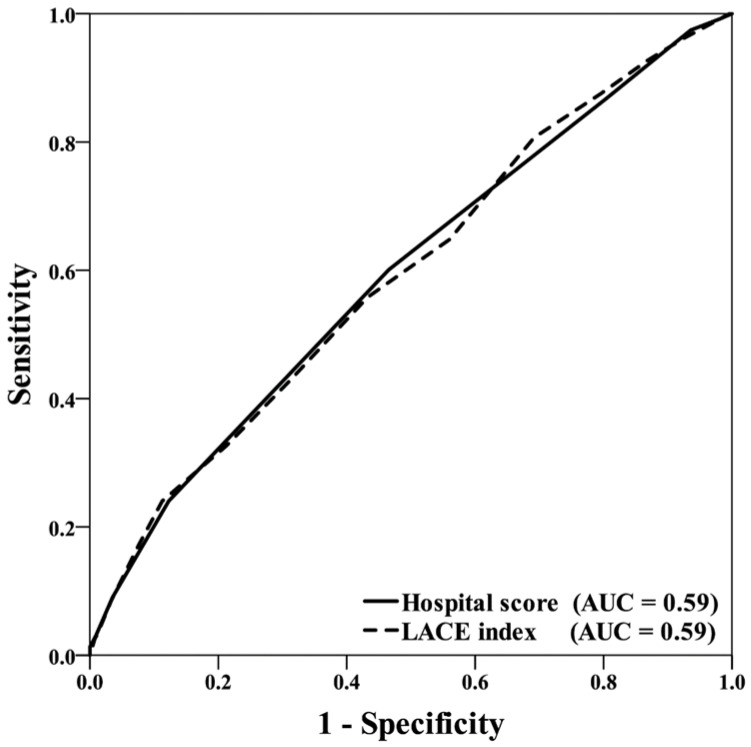
Receiver operator characteristic (ROC) analysis for discrimination ability of the LACE index and HOSPITAL score at predicting 30-day readmissions.

**Table 2 ijerph-20-00348-t002:** Characteristics of the study cohort—1509 patients admitted to acute geriatric ward.

Characteristics	Total (*n* = 1509)	Non-Readmitted (*n* = 1276)	Readmitted (*n* = 233)	*p* Value
Sociodemographic				
Age (years)	82 (75–87)	82 (75–87)	82 (74.5–87)	0.768
Sex, *n* (%)				0.040 *
Male	929 (61.56%)	771 (60.4%)	158 (67.8%)	
Female	580 (38.44%)	505 (39.6%)	75 (32.2%)	
Living situation, *n* (%)				0.846
Alone	137 (9.08%)	117 (9.2%)	20 (8.6%)	
With relatives	1230 (81.51%)	1037 (81.3%)	193 (82.8%)	
Others	142 (9.41%)	122 (9.6%)	20 (8.6%)	
Educational level				0.362
Illiterate	398 (26.38%)	346 (27.1%)	52 (22.3%)	
Literate or primary school	576 (38.17%)	477 (37.4%)	99 (42.5%)	
Junior to senior high school	376 (24.92%)	317 (24.8%)	59 (25.3%)	
University	159 (10.54%)	136 (10.7%)	23 (9.9%)	
Marital status, *n*(%)				0.281
Single	46 (3.05%)	42 (3.3%)	4 (1.7%)	
Married	1463 (96.95%)	1234 (96.7%)	229 (98.3%)	
Age-adjusted Charlson Comorbidity Index (ACCI)	2 (1–3)	2 (1–3)	3 (2–4)	0.005 **
CGA upon admission				
ADL	50 (15–75)	50 (15–75)	40 (10–67.5)	0.005 **
IADL	1 (0–4)	2 (0–4)	1 (0–3)	0.008 **
MMSE	21 (15–26)	21 (15–26)	20 (16–26)	0.895
MNA	21 (16.5–24)	21 (16.5–24.5)	19.5 (16–23.5)	0.005 **
TUG	19.2 (13.6–27.28)	18.8 (13.3–26.83)	20.24 (15.37–29.95)	0.070
Mini-cog	1 (0–1)	1 (0–1)	1 (0–1)	0.279
GDS	1 (0–2)	1 (0–2)	1 (0–3)	0.144
6M	13 (9–20)	12.7 (9–20)	14.3 (8.88–20.19)	0.440
Gait ability				0.001 **
0–1	1029 (68.19%)	894 (70.1%)	135 (57.9%)	
2–3	269 (17.83%)	213 (16.7%)	56 (24.0%)	
4	211 (13.98%)	169 (13.2%)	42 (18.0%)	
Hospital score	4 (4–5)	4 (4–5)	5 (4–5)	<0.001 **
Hospital score > 5	213 (14.12%)	157 (12.3%)	56 (24.0%)	<0.001 **
LACE index	11 (9–13)	11 (9–13)	12 (10–14)	<0.001 **
LACE index > 15	137 (9.08%)	97 (7.6%)	40 (17.2%)	<0.001 **
Length of stay	10 (7–16)	10 (7–15)	11 (8–17)	0.005 **
Length of stay ≥ 14	517 (34.26%)	424 (33.2%)	93 (39.9%)	0.057
N of hospitalization in 6 months prior to index admission				<0.001 **
No	1157 (76.67%)	998 (78.2%)	159 (68.2%)	
1	238 (15.77%)	201 (15.8%)	37 (15.9%)	
2	70 (4.64%)	53 (4.2%)	17 (7.3%)	
≥3	44 (2.92%)	24 (1.9%)	20 (8.6%)	
N of ER visits in 6 months prior to index admission				<0.001 **
No	385 (25.51%)	336 (26.3%)	49 (21.0%)	
1	695 (46.06%)	604 (47.3%)	91 (39.1%)	
2	231 (15.31%)	194 (15.2%)	37 (15.9%)	
3	99 (6.56%)	80 (6.3%)	19 (8.2%)	
≥4	99 (6.56%)	62 (4.9%)	37 (15.9%)	
Laboratory data				
Serum Creatinine	1.02 (0.78–1.5)	1 (0.76–1.48)	1.11 (0.8–1.69)	0.023 *
Serum hemoglobin	11.2 (9.7–12.7)	11.2 (9.7–12.7)	10.9 (9.55–12.65)	0.189
Serum Sodium	139 (136–141)	139 (136–142)	138 (135–141)	0.045 *

Continuous data are expressed as median (IQR); categorical data are expressed as number and percentage. * *p* < 0.05, ** *p* < 0.01.

**Table 3 ijerph-20-00348-t003:** Univariate and multivariate analysis of predictors of 30-day hospital readmissions.

	Univariate			Multivariate (HOSPITAL Score)			Multivariate (LACE Index)		
	OR	95%CI	*p* Value	OR	95%CI	*p* Value	OR	95%CI	*p* Value
Age	1.00	(0.98–1.02)	0.838	0.98	(0.97–1.00)	0.101	0.98	(0.97–1.00)	0.123
Sex									
Male	ref			ref			ref		
Female	0.72	(0.54–0.98)	0.034 *	0.67	(0.49–0.92)	0.012 *	0.70	(0.51–0.95)	0.023 *
Hospital score	1.36	(1.21–1.53)	<0.001 **	1.33	(1.17–1.50)	<0.001 **			
LACE index	1.12	(1.07–1.18)	<0.001 **				1.10	(1.05–1.16)	<0.001 **
Gait ability									
0–1	ref			ref			ref		
2–3	1.74	(1.23–2.46)	0.002 **	1.51	(1.01–2.25)	0.043 *	1.47	(0.99–2.19)	0.057
4	1.65	(1.12–2.41)	0.011 *	1.25	(0.77–2.05)	0.370	1.23	(0.75–2.01)	0.410
Assessment upon admission									
ADL	0.99	(0.99–1.00)	0.005 **	1.00	(0.99–1.01)	0.628	1.00	(0.99–1.01)	0.715
MNA	0.96	(0.94–0.99)	0.006 **	1.00	(0.96–1.04)	0.828	0.99	(0.96–1.03)	0.772

Logistic regression. * *p* < 0.05, ** *p* < 0.01.

**Table 4 ijerph-20-00348-t004:** Sensitivity and specificity for each cut-off point of LACE index and HOSPITAL score at predicting 30-day readmissions.

	Sensitivity	Specificity
LACE index		
3	0.0%	99.6%
4	0.4%	99.2%
5	1.3%	97.6%
6	2.1%	96.0%
7	3.4%	94.4%
8	5.2%	92.8%
9	6.9%	89.6%
10	15.9%	86.8%
11	9.0%	87.2%
12	12.9%	88.0%
13	10.3%	90.0%
14	8.6%	90.0%
15	6.9%	96.3%
16	8.6%	95.8%
17	5.2%	97.9%
18	0.9%	99.5%
19	2.6%	99.2%
Hospital score		
1	0.9%	98.0%
2	1.7%	95.5%
3	10.7%	86.8%
4	26.6%	66.2%
5	36.1%	65.8%
6	15.0%	91.2%
7	6.0%	97.4%
8	1.7%	99.1%
9	0.9%	100%
10	0.4%	100%

**Table 5 ijerph-20-00348-t005:** Subgroup analysis of ROC analysis for discrimination ability of the LACE index and HOSPITAL score at predicting 30-day readmissions.

	HOSPITAL Score			LACE Index		
	AUC	(95%CI)	*p* Value	AUC	(95%CI)	*p* Value
Overall	0.59	(0.55–0.63)	<0.001 **	0.59	(0.55–0.63)	<0.001 **
MNA < 24	0.59	(0.54–0.63)	<0.001 **	0.59	(0.54–0.64)	<0.001 **
MNA ≥ 24	0.60	(0.52–0.67)	0.020 *	0.58	(0.50–0.66)	0.052
ADL (upon admission) < 60	0.59	(0.54–0.64)	<0.001 **	0.60	(0.55–0.65)	<0.001 **
ADL (upon admission) ≥ 60	0.58	(0.51–0.65)	0.024 *	0.56	(0.50–0.63)	0.072
Gait ability 0–1	0.58	(0.52–0.63)	0.003 **	0.57	(0.52–0.63)	0.005 *
Gait ability≥ 2	0.58	(0.51–0.64)	0.019 *	0.58	(0.51–0.64)	0.017 *
Age < 85	0.58	(0.53–0.63)	0.002 **	0.58	(0.53–0.63)	0.002 **
Age ≥ 85	0.62	(0.55–0.68)	0.001 **	0.61	(0.55–0.68)	0.001 **
Male	0.60	(0.55–0.65)	<0.001 **	0.61	(0.56–0.66)	<0.001 **
Female	0.59	(0.51–0.66)	0.017 *	0.55	(0.48–0.63)	0.133

* *p* < 0.05, ** *p* < 0.01.

## Data Availability

Not applicable.

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
