# Peer review of "Prediction of 30-Day Readmission in Hospitalized Older Adults Using Comprehensive Geriatric Assessment and LACE Index and HOSPITAL Score"

_ijerph, 2022, doi:10.3390/ijerph20010348_

Round 1
Reviewer 1 Report
The authors retrospectively investigated the impact of Comprehensive Geriatric Assessment (GCA) / LACE 3 Index / HOSPITAL Score on 30-Day Readmission in Hospitalized Older Adults. The reviewer agrees with the importance of this study but needs to address some critical points to find generalize their findings.
#1 The major reason for hospitalization in patients aged 65 years or older need to show. The key point was whether those diseases are recoverable or not. If those are unrecoverable, those patients need long hospitalization and have a higher rate of rehospitalization.
#2 Definition of discharge needs to address. If patients with poor general status with unrecoverable disease are discharged (such as political reasons), it is not hard to imagine them being readmitted soon. Social background-related factors are problems that cannot be solved by medical care alone, and frailty assessment makes little sense. Authors should focus on clinically important issues.
#3 Looking at the patient background, this target population is a group of patients who are too frail. Frailty assessment of these groups only reveals that the frailty is more severe. Furthermore, rehospitalization does not appear to be an important outcome in this group. Rather than predicting rehospitalization, it would be more medically meaningful to examine what interventions might be appropriate.
#4 The authors use many frailty assessment tools and confusing for non-specialists for frailty assessment. Those should be explained via tables or figures. As no definitive tools are available for CGA, the authors should present clearly what each tool is measuring (ex. GDC-5, NMA, LACE, and HOSPITAL score)
#5 This is the key point. In general, it is difficult for us to conduct CGA in our clinical practice timely. The authors should consider how this could be easily implemented. This is the biggest hurdle to frailty assessment.
Author Response
Dear reviewer,
Thanks for your precious opinions and comments. We have made revisions according to your suggestions. Here is a point-by-point response of your comments and suggestions.
The authors retrospectively investigated the impact of Comprehensive Geriatric Assessment (GCA) / LACE 3 Index / HOSPITAL Score on 30-Day Readmission in Hospitalized Older Adults. The reviewer agrees with the importance of this study but needs to address some critical points to find generalize their findings.
Comments:
#1 The major reason for hospitalization in patients aged 65 years or older need to show. The key point was whether those diseases are recoverable or not. If those are unrecoverable, those patients need long hospitalization and have a higher rate of rehospitalization.
The common etiology of admission in our cohort has been shown in previous report and updated in the Methods section. The patients were all admitted to our ward due to acute illness, and the top leading causes of admission among our population were pneumonia, urinary tract infection and cellulitis. These additional descriptions were in Method section, in page 3 of the manuscript, tracking line 107-112.
#2 Definition of discharge needs to address. If patients with poor general status with unrecoverable disease are discharged (such as political reasons), it is not hard to imagine them being readmitted soon. Social background-related factors are problems that cannot be solved by medical care alone, and frailty assessment makes little sense. Authors should focus on clinically important issues.
The definition of discharge has been described in Method section, in page 2 of the manuscript, tracking line 107-112. As shown in our Methods section, discharge was arranged only after recovery of acute illness such as infection. None of the patients were discharged against medical advice.
#3 Looking at the patient background, this target population is a group of patients who are too frail. Frailty assessment of these groups only reveals that the frailty is more severe. Furthermore, rehospitalization does not appear to be an important outcome in this group. Rather than predicting rehospitalization, it would be more medically meaningful to examine what interventions might be appropriate.
We have reviewed on intervention towards phenotype of frailty and impairment in CGA, in Discussion section, in page 13 of manuscript, tracking line 360-376, though we did not analyze the outcome of intervention among our population during this study. This would be future directions for study.
#4 The authors use many frailty assessment tools and confusing for non-specialists for frailty assessment. Those should be explained via tables or figures. As no definitive tools are available for CGA, the authors should present clearly what each tool is measuring (ex. GDC-5, NMA, LACE, and HOSPITAL score)
The instruments used in CGA has been described in detail in Table 1 in Methods section, page 4 of the manuscript.
#5 This is the key point. In general, it is difficult for us to conduct CGA in our clinical practice timely. The authors should consider how this could be easily implemented. This is the biggest hurdle to frailty assessment.
We acknowledge that CGA is time-consuming and may limit the applicability of our study. However, if properly assessed and tailored treatment being carried out thoroughly afterwards, CGA may still be beneficial and decrease mortality and care dependence in elders with geriatric syndrome and frailty. As described in Discussion section, in page 14 of the manuscript, tracking line 360-376, there have been modified and brief versions of CGA, i.e., Rapid Geriatric Assessment, that was less time-consuming and could screen patients of early geriatric syndrome. Future directions of study may be towards the feasibility of such assessment tools.
We thank you again for your kindly comments and suggestions!
Best regards,
Dr. Chia-Hui Sun
Reviewer 2 Report
Review of “Prediction of 30-day readmission…”
30-day readmissions are a significant problem in healthcare. In the US, depending on diagnosis, the 30-day readmission rate can be in the range of 15-20%. US Centers for Medicare and Medicaid Services some years ago instituted the Hospital Readmission Reduction Program (HRRP) to penalize hospitals with high readmission rates, a program that has somewhat reduced the readmission rates.
One tool for predicting (at admission) which patients are likely to be readmitted is the LACE index. Originally developed in a Canadian hospital it has become widely-used because it is both simple (4 factors) and uses data available at admission without special testing. In their current paper the authors compare more sophisticated methods with the LACE index and find (unsurprisingly) that multi-variable methods are more accurate predictors of readmission. Indeed the number of variables recorded in the present study is large and one is inclined to lose track. I also wonder how long the administration of the various tests takes at admission and the resources required (LACE, for all its faults, is simple, quick and requires minimal resources).
The other algorithm that is tested is the HOSPITAL score, with which I am not familiar. This score is based on 7 variables with some overlap with LACE, as well as some clinical values. I expect that calculation of this score would require more resources and take longer than LACE.
In terms of patients, those with readmissions had more prior admissions and ER visits, more co-morbidities and performed worse on CGA measures. Both LACE and HOSPITAL models performed poorly on the basis of their ROC curves (not surprisingly). I expected that the contribution of this paper would be to identify those components of the CGA that are predictive of readmission, in order to construct a more refined test than LACE to apply at admission. To this end Table 2 should provide some answers. However, I am confused by the different models reported in Table 2: I understand univariate analysis but what do the 4 models represent? Although I searched I could not find a reference to them in the paper. I also would have liked to have seen a straightforward discussion of the CGA factors that are predictive; the authors throw in detailed discussion of other studies into the discussion of the results, which distracts from understanding what is significant in this study. It appears that there are few factors (in addition to those included in LACE or HOSPITAL) that are predictive (gait and male sex) but I may not be interpreting the results correctly.
I have made some references to resources in this review. If the authors are correct that there are few factors that are predictive (in addition to LACE or HOSPITAL) then resources at admission will not be an issue. The other resource issue, however, is what can be done for patients to reduce the readmission risk? One factor that appears to be missing from this study is the presence and quality of post-discharge resources. The US HRRP program has increased the prevalence of post-discharge programs as one solution to the readmission problem. Depending on the availability of these programs it is possible that a less-accurate model, requiring less testing at admission, will be satisfactory in terms of identifying candidates for these programs. [1] is a paper that includes a theoretical economic model of post-discharge care.
1. Duncan I. Huynh N., A Predictive Model for Readmissions Among Medicare Patients in a California Hospital. Population Health Management, 2018. 21: p. 317-22.
Author Response
Dear reviewer,
Thanks for your precious opinions and comments. We have made revisions according to your suggestions. Here is a point-by-point response of your comments and suggestions.
Review of “Prediction of 30-day readmission…”
30-day readmissions are a significant problem in healthcare. In the US, depending on diagnosis, the 30-day readmission rate can be in the range of 15-20%. US Centers for Medicare and Medicaid Services some years ago instituted the Hospital Readmission Reduction Program (HRRP) to penalize hospitals with high readmission rates, a program that has somewhat reduced the readmission rates.
1.One tool for predicting (at admission) which patients are likely to be readmitted is the LACE index. Originally developed in a Canadian hospital it has become widely-used because it is both simple (4 factors) and uses data available at admission without special testing. In their current paper the authors compare more sophisticated methods with the LACE index and find (unsurprisingly) that multi-variable methods are more accurate predictors of readmission. Indeed the number of variables recorded in the present study is large and one is inclined to lose track. I also wonder how long the administration of the various tests takes at admission and the resources required (LACE, for all its faults, is simple, quick and requires minimal resources)..The other algorithm that is tested is the HOSPITAL score, with which I am not familiar. This score is based on 7 variables with some overlap with LACE, as well as some clinical values. I expect that calculation of this score would require more resources and take longer than LACE.
The CGA was recorded and assessed by practitioners as well as well-trained nurses in our facility. Unfortunately, the administration time for each CGA was not recorded in our study. We do acknowledge that CGA is time-consuming and may limit the applicability of our study; however, if properly assessed and tailored treatment being carried out thoroughly afterwards, CGA may still be beneficial and decrease mortality and care dependence in elders with geriatric syndrome and frailty (as described in Discussion section, in page 14 of the manuscript, tracking line 360-376).
2.In terms of patients, those with readmissions had more prior admissions and ER visits, more co-morbidities and performed worse on CGA measures. Both LACE and HOSPITAL models performed poorly on the basis of their ROC curves (not surprisingly). I expected that the contribution of this paper would be to identify those components of the CGA that are predictive of readmission, in order to construct a more refined test than LACE to apply at admission. To this end Table 2 should provide some answers. However, I am confused by the different models reported in Table 2: I understand univariate analysis but what do the 4 models represent? Although I searched I could not find a reference to them in the paper. I also would have liked to have seen a straightforward discussion of the CGA factors that are predictive; the authors throw in detailed discussion of other studies into the discussion of the results, which distracts from understanding what is significant in this study. It appears that there are few factors (in addition to those included in LACE or HOSPITAL) that are predictive (gait and male sex) but I may not be interpreting the results correctly.
We have made an adjustment of Table 3, in Results section, in page 9 of 21 with detailed description in Results section, page manuscript tracking line 192-200, where LACE index and HOSPITAL score were included in multivariable regression analysis respectively after univariable regression.
We have also added discussion of the significant positive findings of this study (gait, male sex) in Discussion section, in page 12 of the manuscript, tracking line 284-302.
3.I have made some references to resources in this review. If the authors are correct that there are few factors that are predictive (in addition to LACE or HOSPITAL) then resources at admission will not be an issue. The other resource issue, however, is what can be done for patients to reduce the readmission risk? One factor that appears to be missing from this study is the presence and quality of post-discharge resources. The US HRRP program has increased the prevalence of post-discharge programs as one solution to the readmission problem. Depending on the availability of these programs it is possible that a less-accurate model, requiring less testing at admission, will be satisfactory in terms of identifying candidates for these programs. [1] is a paper that includes a theoretical economic model of post-discharge care.
We have reviewed on intervention towards phenotype of frailty and impairment in CGA, in Discussion section, in page 13 of manuscript, tracking line 360-376, though we did not analyze the outcome of intervention among our population during this study. This would be future directions for study.
We thank you again for your kindly comments and suggestions!
Best regards,
Dr. Chia-Hui Sun
Reviewer 3 Report
This is an interesting paper which investigated potential predictors of 30-day readmission risk by Comprehensive Geriatric Assessment (CGA) in hospitalized patients aged 65 years or older, and examined the predictive ability of LACE index and HOSPITAL score in older patients with combination of malnutrition and physical dysfunction. It has clear contribution to current knowledge.
However, the document needs to be re-structured a bit, clarify the results section, and focus the discussion according to the objectives of the study.
Other detailed comments are below, I hope, will be helpful as you work on this paper.:
Results:
1. Table 3 & 4 can be improved so that it’s easy to read.
Discussion:
1. The current discussion section mostly repeats the result section and compare it with findings previous studies. The authors need to provide deeper discussion about the main findings of this study.
2. The contribution of this study and the significance of this paper's findings for research, policy, etc., should be better developed.
Author Response
Dear reviewer,
Thanks for your precious opinions and comments. We have made revisions according to your suggestions. Here is a point-by-point response of your comments and suggestions.
Results:
- Table 3 & 4 can be improved so that it’s easy to read.
We have adjusted the description and form of Table 3 & 4 (currently Table 4 & 5) into clearer and more concise form in Results section, in page 9-10.
Discussion:
1. The current discussion section mostly repeats the result section and compare it with findings previous studies. The authors need to provide deeper discussion about the main findings of this study.
We have added discussion of the significant positive findings of this study (gait, male sex) in Discussion section, in page 12 of the manuscript, tracking line 284-302.
2. The contribution of this study and the significance of this paper's findings for research, policy, etc., should be better developed.
As described in Discussion section, in page 14 of the manuscript, tracking line 360-376, we believe that although CGA has been known to be time-consuming and there is difficulty for standardization of assessment, if properly assessed and tailored treatment being carried out thoroughly afterwards, CGA may still be beneficial and decrease mortality and care dependence in elders with geriatric syndrome and frailty. There have also been modified and brief versions of CGA, i.e., Rapid Geriatric Assessment, that was less time-consuming and could screen patients of early geriatric syndrome. Future directions of study may be towards the feasibility of such assessment tools.
We thank you again for your kindly comments and suggestions!
Best regards,
Dr. Chia-Hui Sun